# High measles and rubella vaccine coverage and seroprevalence among Zambian children participating in a measles and rubella supplementary immunization activity

Christine Prosperi[1], Shaun Truelove[1,2]*, Andrea C. Carcelen[1], Gershom Chongwe[3], Francis D. Mwansa[4], Phillimon Ndubani[5], Edgar Simulundu[5], Innocent C. Bwalya[3], Mutinta Hamahuwa[5], Kelvin Kapungu[3], Kalumbu H. Matakala[5], Gloria Musukwa[5], Irene Mutale[3], Evans Betha[3], Nchimunya Chaavwa[5], Lombe Kampamba[3], Japhet Matoba[5], Passwell Munachoonga[5], Webster Mufwambi[3], Ken Situtu[3], Philip E. Thuma[5], Constance Sakala[4], Princess Kayeye[4], Amy K. Winter[2,6,7], Matthew J. Ferrari[8], William J. Moss[1,2], Simon Mutembo[1]

1 Department of International Health, International Vaccine Access Center, Johns Hopkins Bloomberg School of Public Health, Baltimore, Maryland, United States of America, 2 Department of Epidemiology, Johns Hopkins Bloomberg School of Public Health, Baltimore, Maryland, United States of America, 3 Tropical Diseases Research Center, Ndola, Zambia, 4 Government of the Republic of Zambia, Ministry of Health, Lusaka, Zambia, 5 Macha Research Trust, Choma, Zambia, 6 Department of Epidemiology and Biostatistics, University of Georgia, Athens, Georgia, United States of America, 7 Center for the Ecology of Infectious Diseases, University of Georgia, Athens, Georgia, United States of America, 8 Center for Infectious Disease Dynamics, Pennsylvania State University, State College, Pennsylvania, United States of America

☯ These authors contributed equally to this work.
* shauntruelove@jhu.edu

## Abstract

Post-campaign coverage surveys estimate the proportion of previously unvaccinated children vaccinated during supplementary immunization activities (SIA) but cannot address whether susceptible children seronegative to measles and rubella viruses were vaccinated during the SIA. We conducted a cross-sectional serosurvey nested within the November 2020 measles-rubella SIA in Zambia, systematically sampling children aged 9 months to 5 years at 30 SIA sites in Choma and Ndola Districts. A questionnaire administered to caregivers collected the child's demographic information and vaccination history. Dried blood spot specimens were collected from child participants and for Immunoglobulin-G antibodies to measles and rubella viruses by enzyme immunoassay. We used the expected vaccination probability by age derived from Demographic and Health Surveys to quantify the value of the immunization campaign, estimating the proportion of children who would not have otherwise received routine MR doses. Among children enrolled with MR vaccination data (N = 2,364), 2,214 (94%) reported at least one routine MR dose before the SIA. We estimate 5.0% [95%CI, 4.2-5.9%] (n = 118/2364) of children would not have otherwise received routine MR dose 1 without the SIA and 23.4% [95%CI, 21.7-25.1]

**Data availability statement:** The individual survey data were collected under data sharing agreements from Zambia Ministry of Health and the Zambia National Health Research Authority. As per the Zambia Health Research Act, access to data requires approval from the Zambian National Health Regulatory Authority. To obtain this access, please contact Dr Victor Chalwe, Acting Director of the Zambia National Health Research Authority (victor.chalwe@nhra. org.zm).

**Funding:** This study was supported by the Strengthening Immunization Systems through Serosurveillance grant (Grant number 1094816) from The Bill & Melinda Gates Foundation to the International Vaccine Access Center, Department of International Health, Johns Hopkins Bloomberg School of Public Health. CP, ST, ACC, GC, FDM, PN, ES, ICB, MH, KK, KHM, GM, IM, EB, NC, LK, JM, PM, WM, KS, PET, CS, PK, AKW, MJF, WJM, and SM were all supported by grant 1094816 for this study. The funders had no role in study design, data collection and analysis, decision to publish, or preparation of the manuscript.

**Competing interests:** The authors have declared that no competing interests exist.

(n = 553/2364) would not have otherwise received routine MR dose 2. Thus, 1 in 3.5 doses were given to an MR un- or under-vaccinated child who may not have received that dose in the absence of an SIA. Eighty-six percent and 90% of children were measles and rubella seropositive before vaccination during the SIA. Thirty-six percent of children with no prior routine MR dose were measles seropositive, while nine percent of children with two prior routine MR doses were measles seronegative. Although children vaccinated during this SIA were highly likely to have previously received routine vaccinations, the SIA reached a considerable number of susceptible children. Monitoring SIA effectiveness and efficiency through standardized metrics and formats is critical for impactful vaccine delivery strategies.

## Introduction

Despite widespread measles vaccination, measles outbreaks and endemic measles virus transmission continue, resulting in an estimated 9.2 million cases and 136,000 deaths globally in 2022 [1]. To address the challenge of reaching every child with measles vaccine, many countries rely on national supplemental immunization activities (SIAs), mass vaccination campaigns that target all individuals within a defined age range regardless of prior vaccination status. However, measles and rubella SIAs may largely provide additional doses to children already vaccinated through the routine immunization program, particularly in countries with high routine immunization coverage [2]. Children who are not reached by routine immunization services and are unvaccinated against measles, referred to as measles zero-dose children, comprise a vulnerable population who must be reached during an SIA. In addition, 5–15% of children do not develop protective immunity following receipt of their first dose of measles-containing vaccine (MCV) [3–5]. This population of children who either do not receive measles vaccine through the routine immunization system or do not develop protective immunity following vaccination form the pool of susceptible children.

Little is known about the proportion of susceptible children vaccinated during MR SIAs. Post-campaign coverage surveys, conducted shortly after an SIA, estimate the number of zero-dose, previously unvaccinated children vaccinated during the SIA but their results are often subject to bias, substantial uncertainty, and cannot address whether the SIA vaccinated susceptible children who were seronegative to measles and rubella viruses [6]. Understanding whether susceptible children are vaccinated during the SIA can demonstrate the added value of SIAs in closing population immunity gaps.

Serosurveys identify seronegative children and highlight population immunity gaps [7,8]. However, they can be expensive and logistically challenging to conduct [9,10], especially when designed as community-based serosurveys with substantial personnel and transportation costs. Conducting such surveys can be even more difficult in resource-limited settings. Vaccine delivery platforms such as SIAs establish centralized locations where children in the age group of interest for MR serosurveys

present, reducing the resources required to enroll children but acknowledging that children not participating in the SIA are not included in the serosurvey. To our knowledge, there is no published literature on measles and rubella serosurveys nested within an SIA, although integrated delivery of other interventions, such as vitamin A supplementation and deworming, is common and has not been shown to negatively impact vaccine coverage [11].

Zambia has a strong childhood vaccination program with 93% coverage in 2019 for the first dose of measles-rubella (MR) vaccine but only 66% for the second dose [12]. The Zambian Ministry of Health also has a long history of successful SIAs, reaching over 3 million children with MR vaccines during Child Health Week in November 2020 [13]. However, measles outbreaks subsequently were reported in several districts in 2022 and 2023, possibly due to lower than optimal coverage of the November 2020 MR vaccination campaign and disruptions to the routine immunization program in 2020–2021 due to the COVID-19 pandemic [14,15]. Since the impact of SIAs on population immunity cannot be adequately determined by vaccination coverage estimates alone, we conducted a MR serosurvey nested within the SIA to estimate the proportion of susceptible children vaccinated during the November 2020 MR SIA in Zambia.

## Materials and methods

### Ethics statement

Ethical approval was obtained from the Tropical Diseases Research Center Ethics Review Committee, and Johns Hopkins Institutional Review Board. Further regulatory approval was provided by the Zambia National Health Research Authority. Written informed consent was obtained from the parent or guardian of each child before enrollment in the study. Additional information regarding the ethical, cultural, and scientific considerations specific to inclusivity in global research is included in the Supporting Information (S1 Checklist).

### Overview of study design and setting

We conducted a descriptive cross-sectional serosurvey nested within the November 2020 MR SIA that was part of Child Health Week activities in Zambia. The serosurvey was conducted in two districts, Ndola District, Copperbelt Province in the northern zone of the country, which is primarily urban and had 87% routine MCV1 coverage in 2019, and Choma District, Southern Province in the southern zone of the country, which is primarily rural and had 90% routine MCV1 coverage in 2019.

### Survey setting

In each district, we purposefully selected 10 MR SIA vaccination sites. The selected sites included a mix of rural health centers, urban health centers, hospital-affiliated health center, and health posts. Health Facility- and community-level characteristics were obtained for each facility with input from the site staff. These characteristics included geographic location and setting (urban versus rural), type of health facility, size of the health facility catchment area, functionality of the health facility based on national performance metrics, and details on accessibility of the health facility or difficult to reach subpopulations in the catchment area. The study team manually reviewed the list of health facilities and characteristics to purposefully select 10 health facilities per district to capture the range of characteristics. Five facilities in each district were selected to have serosurvey enrollment at both fixed and outreach posts, for a total of 30 campaign sites between the two districts. Five facilities in each district were selected to have serosurvey enrollment at both fixed and outreach posts, for a total of 30 campaign sites between the two districts. Outreach sites expected to have the most children during the campaign were selected. If the outreach site varied by day during the campaign for a given facility, the survey teams moved with the vaccination team to the different locations where possible. Decisions regarding selection of outreach sites and movement of survey teams were made by the central study team based on the campaign microplan.

The serosurvey teams were embedded in the campaign activities at each campaign site and recruited children for the serosurvey immediately following vaccination. Children aged 9 months to 5 years who were vaccinated at the selected campaign sites were systematically sampled for the serosurvey, with sampling designed to enroll a similar number of participants at each health facility and distribute enrollment across the days of the campaign [16]. Daily enrollment caps and sampling intervals were set based on the SIA microplan and the number of children expected to be vaccinated per day. Children were enrolled in the study over the six days to capture variation in the population and vaccination status. The estimated sample size was 1,200 children in each district (maximum of 2,400 children), based on an expected measles seroprevalence of 80% and 5% precision, with a type I error of 0.05, design effect of 2.1 and non-response rate of 35%.

The survey team approached families of children who were systematically sampled after vaccination to introduce the serosurvey and obtain parental permission. After obtaining written informed consent from the parent or guardian of each child, demographic and vaccination history data were collected using a standardized questionnaire on a tablet-based application (REDCap Mobile). Data were uploaded daily during the campaign and reviewed to provide near real-time feedback on data quality concerns and to inform the systematic sampling procedures for the next day. The recruitment period for the study was from November 23 to November 29, 2020.

## Specimen collection and testing

Blood was collected by finger prick using a retractable lancet and spotted onto Whatman 903 filter paper as a dried blood spot (DBS) by trained survey staff. DBS specimens were dried for a minimum of 15–30 minutes then wrapped in a sealed plastic bag with a desiccant. Specimens were dried overnight, repackaged with desiccants in sealed plastic bags, and then stored at -20 °C at the research institute's laboratory until testing.

Dried blood spot samples were eluted according to the manufacturer's protocol and tested for anti-measles virus and anti-rubella virus IgG antibodies using a commercial enzyme linked immunoassay (Euroimmun, Perkin Elmer, Germany) following manufacturer recommendations. Equivocal samples were retested and the repeat result was treated as final. See Supplemental Methods (S1 Text) for additional details on testing procedures.

A stratified random subsample of 300 DBS specimens was selected from those collected in Ndola District and shipped to the Centers for Disease Control and Prevention (CDC) laboratory in Atlanta, Georgia, USA and tested in the CDC's Viral Preventive Diseases Branch using their multiplex bead assay (MBA) [17]. An adjustment to the measles quantitative results was calculated by comparing the MBA and EIA results then applied to the measles quantitative results for all specimens tested by EIA (see S1 Text). For the primary analysis we used the measles MBA threshold (153 mIU/ml) to define measles seropositivity [17,18].

## Statistical analyses

We described the demographic characteristics of enrolled children then summarized vaccination status based on the number of routine MR vaccine doses received prior to the SIA according to the vaccination card or caregiver recall if the card was unavailable. Children with unknown vaccination receipt based on recall were excluded from vaccination-related analyses. Estimates of measles-rubella vaccine dose 1 (MR1) and dose 2 (MR2) vaccination status among children 12–23 months of age and 24–35 months of age, respectively, were compared with 2020 estimates from WHO-UNICEF (WUENIC) [12]. Associations between vaccination status and day of the SIA and other child-level characteristics, including urban/rural, fixed/outreach vaccination site, travel time to the campaign, vaccination card availability, number of siblings below 5y, missing DTP1, and child's age, were explored using logistic regression. Predicted probability of MR1 at 12 months, or MR2 at 24 months, was estimated for each of the 30 SIA sites using district-specific logistic regression models adjusted for age in years to account for differences in age distribution by SIA site (see S1 Text). Measles and rubella seroprevalence were estimated with 95% Wald confidence intervals. Associations between measles seropositivity and child-level characteristics were explored using logistic regression. Analyses were conducted using R (version 4.2.1).

## Effectiveness and efficiency of vaccination activities

To assess ability of the SIA to reach children that needed vaccination, we defined two novel metrics, the *vaccination activity effectiveness (VAET)* and *vaccination activity efficiency (VAEC)*. V*accine activity effectiveness* represents the ability of the SIA to proportionately reach MR un- and under-vaccinated children in the population, as compared to a random selection of individuals in that population, in other words, does the activity do better, the same, or worse than random sampling. *VAET* is defined as an odds ratio (OR), ranging from 0 to infinity, calculated as the ratio of the odds of a child included in the vaccination activity having received a routine dose to the odds of a child in the general population having received the same routine dose. A *VAET* = 1 can be interpreted as the SIA capturing a random selection of the population targeted by the SIA. As the *VAET* moves from 1 to 0, it indicates the effectiveness of the SIA to reach unvaccinated children is declining to be worse than a random sample, indicating the SIA is oversampling from previously vaccinated children. As the *VAET* moves from 1 to infinity, vaccine activity effectiveness improves, indicating the SIA is reaching more un- or under-vaccinated children than a random selection.

V*accine activity efficiency*, or *VAEC*, represents how many doses it takes the SIA to capture a single un- or under-vaccinated child who would not have otherwise received routine MR doses, adjusting for age and routine vaccination in that population. It is defined as a proportion, ranging from 0 to 1, and is calculated as the total number of administered SIA doses that went to children who otherwise would never have received each routine MR dose, divided by the total number of administered SIA doses. For this metric, we assume age-specific probabilities receiving each dose (i.e., probability a child has received MR1 by 9, 10, 11, etc. months of age) following those captured by the 2018 Demographic and Health Survey data for each district [19]. We then used these to estimate the probability that a child will still be vaccinated, assuming they have not yet by each age, what we are calling the "hazard of vaccination", or *HoV*. We then assign a *HoV* to each individual in the study population given their age; children who already received the dose are assigned a probability of 1. Averaging these probabilities, we get an adjusted estimate of the eventual vaccination coverage via routine systems in this population, and one minus this mean hazard of vaccination gives us the *vaccine activity efficiency;* we also can express this metric as 1 child reached per N doses given. A *VAEC* of 1 is the best possible value, indicating the vaccination activity reached a child who truly needed the vaccine with every dose given; in contrast a very low *VAEC* indicates an inefficient activity: *VAEC* = 0.05 indicates it required 20 doses to reach 1 child who truly needed the vaccine. See S1 Text for more information.

## Results

Over the course of six days, 2,942 children aged 9 months to 5 years were approached for enrollment from approximately 25,088 children who received measles and rubella vaccination across the 30 serosurvey study sites. Of those approached 2,400 (82%) were enrolled, with 1203 and 1197 enrolled from Ndola and Choma Districts, respectively (Table 1). Only 75% of children approached at the SIA sites in Choma District agreed to participate compared to 91% in Ndola District, resulting in a median of 191 and 206 enrolled daily over the six days in Choma and Ndola Districts (range: 53–377 in Choma, 151–269 in Ndola).

Enrolled children had a mean age of 2.6 years (interquartile range [IQR]: 1.5, 3.6) and 51% were male. Forty percent of caregivers had secondary or higher education in Choma District compared to 66% in Ndola District. Three-quarters of children enrolled in Ndola District reached the SIA site within 30 minutes, compared to only half in Choma District, consistent with a more rural setting.

## Vaccination status

Of the 2,400 children enrolled in the serosurvey, 2,171 (90%) had an under-5 vaccination card. Receipt of BCG and at least 1 DTP dose by under-5 card or parental recall was 97% and 98%, respectively, similar to the 97% coverage

**Table 1. Characteristics of children enrolled in a measles and rubella serosurvey nested within a measles and rubella supplementary immunization activity in Zambia, November 2020.**

| | Choma District (N = 1197) | Ndola District (N = 1203) |
|---|---|---|
| Enrolled/approached | 1197/1606 (74.5%) | 1203/1326 (90.7%) |
| Median age, years (IQR) | 2.6 (1.7, 3.7) | 2.3 (1.5, 3.5) |
| Maternal education | | |
| Secondary or higher | 477 (39.8%) | 798 (66.3%) |
| Primary or less | 718 (60.0%) | 404 (33.6%) |
| Travel time to campaign site | | |
| < 30 minutes | 601 (50.2%) | 921 (76.6%) |
| > 30 minutes | 578 (48.3%) | 280 (23.3%) |
| Routine immunization card availability | 1064 (88.9%) | 1107 (92.0%) |
| At least 1 dose of DTP | 1159 (96.8%) | 1183 (98.3%) |
| BCG receipt | 1155 (96.5%) | 1167 (97.0%) |

A serosurvey was nested in a Child Health Week in Zambia in two districts in Zambia, Choma and Ndola. During this serosurvey, 2,400 children were enrolled, surveyed, and had blood taken to measure seropositivity to measles and rubella viruses.

estimates for the respective provinces from the Zambian Demographic and Health Survey (DHS) [19]. Thirty-six children (1.5%) had unknown vaccination status and were treated as missing data and excluded from vaccination-related analyses. Twenty-two percent of caregivers reported their child's routine vaccinations were delayed due to the COVID-19 pandemic.

Among children enrolled with MR vaccination data (n = 2,364; 98.5%), 94% (n = 2,214) reported at least one MR dose before the SIA, either via card (n = 2,012; 91%) or recalled by their caregiver (n = 202; 9%). Children reporting no MR doses prior to the SIA (n = 150) were proportionally dominated by those just eligible for routine receipt, with the proportion of children with no prior MR doses decreasing with age: 44% (83/187) of children 9–11 months, 5% (35/735) of those 12–23 months, and 2% (32/1442) of those older than 23 months had no prior MR dose (Fig 1). The median age of routine MR1 receipt was 9.5 months [IQR: 9.2, 10], while the median age of children receiving MR1 through the SIA was 11 months (IQR: 9.6, 20.8) (S1 and S2 Figs). Additionally, 62% (1,456/2,364) reported two MR doses prior to the campaign, either via card (n = 1292; 89%) or recalled by their caregiver (n = 164; 11%). Similar to MR1, the proportion of children missing the second dose decreased with age, with 94% (421/446) of children aged 9–17 months, 38% (137/358) of children 18–23 months, 14% (80/554) of children 24–35 months, and 14% (120/856) of children older than 35 months not having received MR2. The median age of routine MR2 receipt was 18.7 months (IQR 18.3, 19.3), while the median age of children receiving MR2 through the SIA was 17.2 months (IQR: 13.6, 24.8) (S1 and S2 Figs).

Of children 12–23 months, 95% received at least 1 dose of MR vaccine prior to the SIA whereas 84% of children 24–35 months received two doses of MR prior to the SIA. This MR1 coverage of 95% among surveyed children attending the SIA was similar to the national estimate in 2020 (96%); however, the MR2 estimate of 84% was substantially higher than the 74% national estimate [12]. Based on these prior coverage estimates, we estimated the SIA to have had a moderate *vaccination activity effectiveness* of 1.53, indicating inclusion in the SIA was slightly better than a random sample of the population in terms of reaching unvaccinated children. However, for MR2, we identified a high correlation between routine MR2 vaccination receipt and SIA participation: children who had already received MR2 through the routine system were almost twice as likely to attend the SIA than those who had not received MR2 (odds ratio [OR]=1.84). This translates to a low *vaccination activity effectiveness* of 0.54 for MR2 (i.e., worse than a random sample).

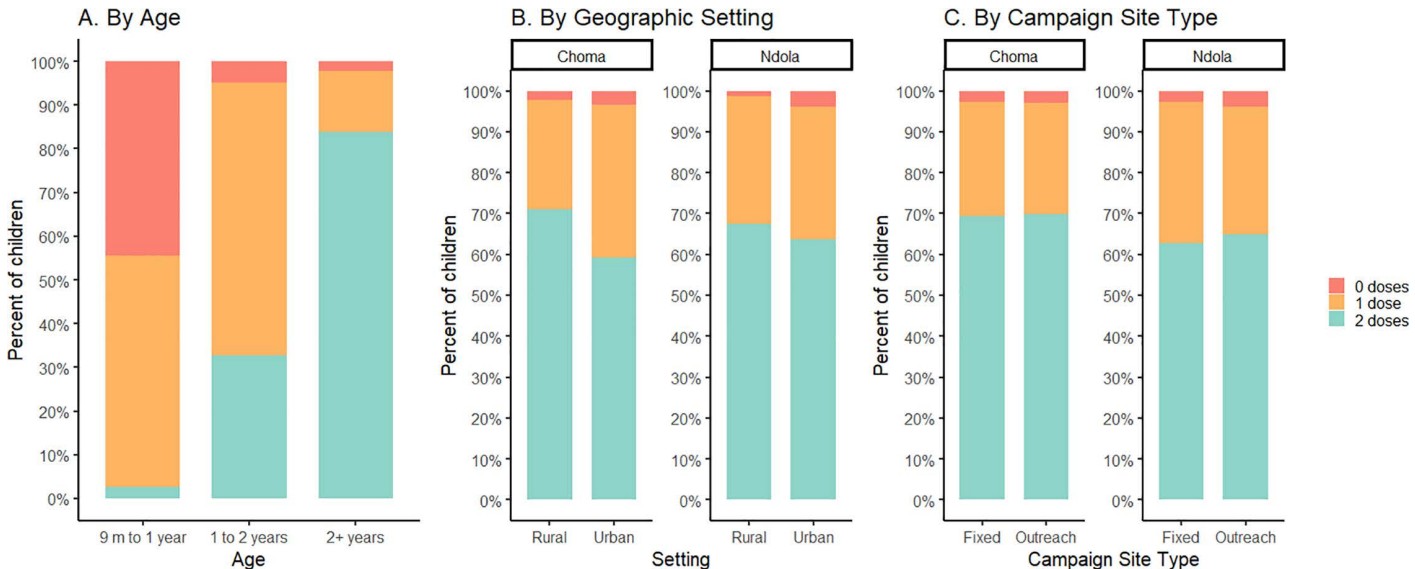

**Fig 1. Number of measles-containing vaccine doses received prior to the SIA, Choma and Ndola, Zambia, November 2020.** The analysis was stratified by age, geographic setting, and campaign site type, and was restricted to children >1 year. SIA type was restricted to health facilities with both fixed and outreach locations.

Through hazard analysis and assuming the probability of vaccination by age estimated from the 2018 DHS, we estimate that without the SIA, 22% (95% confidence interval [CI], 18–25%) of children with zero doses prior to the SIA would have eventually received a first routine dose, and 39% (95% CI, 34–40%) of children with one dose would have eventually received a second routine dose, through the routine immunization system. With these adjustments, we estimate 95% (95% CI, 94.8-95.2%) eventual MR1 coverage and 77% (95% CI, 75–77%) MR2 coverage through routine immunization among the study children seeking vaccination at the SIA. This suggests the SIA still reached children with doses who would not have received them otherwise: we estimate 5.0%, or 118 of 2,364 children, would not have otherwise received MR1 without the SIA, and 23.4% (553/2364) would not have otherwise received MR2. This equates to a first dose *vaccination activity efficiency* of 0.05, or 1 measles unvaccinated child reached per 20 children vaccinated by the SIA and second dose *vaccination activity efficiency* of 0.23, or 1 under-vaccinated child in 4.3 children vaccinated in the SIA, and an overall *vaccination activity efficiency* of 0.28, or 1 out of every 3.5 SIA doses being given to a child who otherwise would not have gotten it.

There was minimal difference by district in the percentage of children with prior MR receipt (Ndola: 92% and Choma: 95%). There was no difference in prior MR receipt by rural versus urban setting or type of SIA site (fixed versus outreach) in either district (Fig 1, S1 Table). Those missing BCG or DTP1 were more likely to be measles zero-dose at time of the SIA but this was only significant for DTP1 in Ndola District (OR [95% CI], 7.9 [1.7, 26.9]) (S1 Table). Children 24 months or older attending an outreach SIA site in Ndola District were almost twice as likely to be missing MR2 as those attending fixed sites (OR [95% CI], 1.8 [1.0, 3.3]) (S2 Table). In both districts, having two or more siblings under 5 years of age was associated with missing MR2 (OR [95% CI], Choma: 3.4 [1.1, 9.1]; Ndola: 4.0 [1.4, 10.1]. Lower maternal education was also associated with missing MR2, though only significant in Choma District (OR [95% CI], 1.9 [1.2, 3.0]).

Within each district there was variability in the number of MR doses received by SIA site (Fig 2A), although the age of the children at each site also varied (median age range 2.0 to 3.4 years, after excluding children 9–11 months). After adjusting for age, the predicted probability of a child having received MR1 or MR2 at a fixed age (12 months and 24 months, respectively) continued to vary by SIA site (Fig 2B).

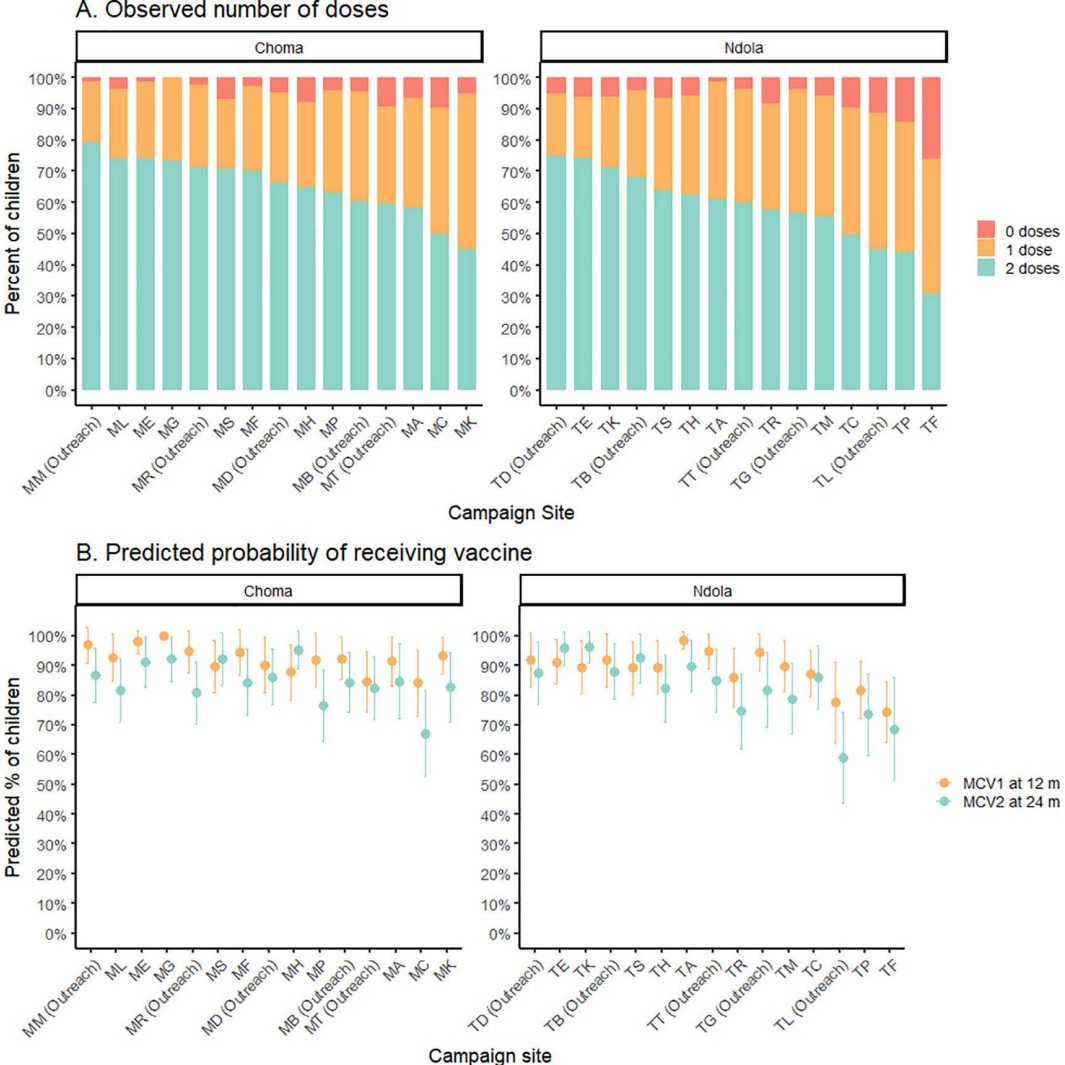

**Fig 2. Variability in MR routine vaccination status by campaign site, observed and predicted, among children participating in the November 2020 SIA in Zambia.** (A) Observed number of doses received prior to the 2020 SIA in Choma and Ndola Districts, Zambia. (B) Predicted probability of receiving MR1 at 12 months and MR2 at 24 months. We estimated district-specific predicted probability of a child 12 months of age having received MR1 or a child 24 months of age having received MR2 using a logistic regression model adjusted for age and campaign site. The model for MR2 was restricted to children 24 months and older.

### Variation by day of the SIA

Characteristics of the children enrolled varied over time, with substantial variation occurring at the district level by day of enrollment (S3 Fig). The average age significantly increased over time in both districts, ranging from 50% of children older than 24 months on the first day to 69% on the last day. We observed variability by day in the number of siblings, maternal education, and sex of the children attending the SIA in Choma District. Children with no siblings younger than 5 years of age and mothers with higher levels of education were more commonly observed at the beginning and end of the SIA in Choma District (S3 Fig). Reported travel time to reach the SIA site varied by day of the campaign in both districts. There was no association in the percentage of children with a vaccination card or those having received other routine immunizations (DTP1 or BCG).

PLOS Global Public Health

Since there was little variability in MR receipt by district, we combined the data to examine variability in MR receipt by day. After stratifying by age group, there was no difference by day in the percent of children with zero MR doses prior to the SIA except for children 12–23 months of age (Fig 3A–C). Restricting to children 24 months and older with at least one prior MR dose, there was no difference in the percentage receiving their second dose through the SIA by day (Fig 3D). In outreach sites, the percentage of zero-dose children was highest on the last day of the SIA (S4 Fig).

## Measles and rubella seroprevalence

Overall, 2,061 (86%) children were measles seropositive and 2,161 (90%) children were rubella seropositive before vaccination during the SIA. Measles seroprevalence was similar between the two sites (Fig 4A). Measles seroprevalence was 58% (95% CI: 51–65%) among those 9 months to <1 year, 88% (95% CI: 86%-91%) among those 1–2 years, and 88%

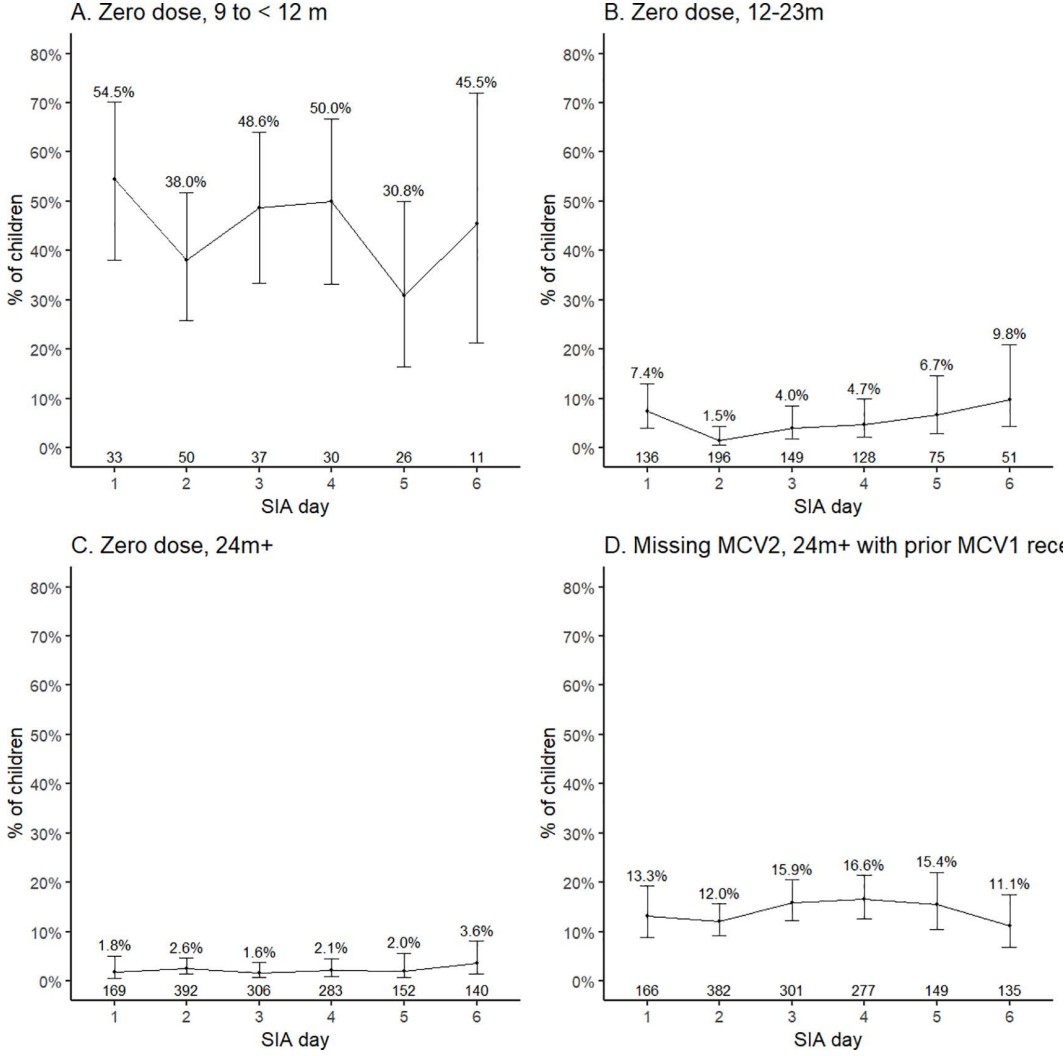

**Fig 3. Variability in vaccination status of children reached through the SIA by day and age group, November 2020, Zambia.** We estimated the proportion of children attending the SIA who were measles zero-dose stratified by day of attendance and age group. A similar analysis was performed to estimate the proportion missing the second dose of MR among children 2 years and older who had received at least one dose prior to the SIA.

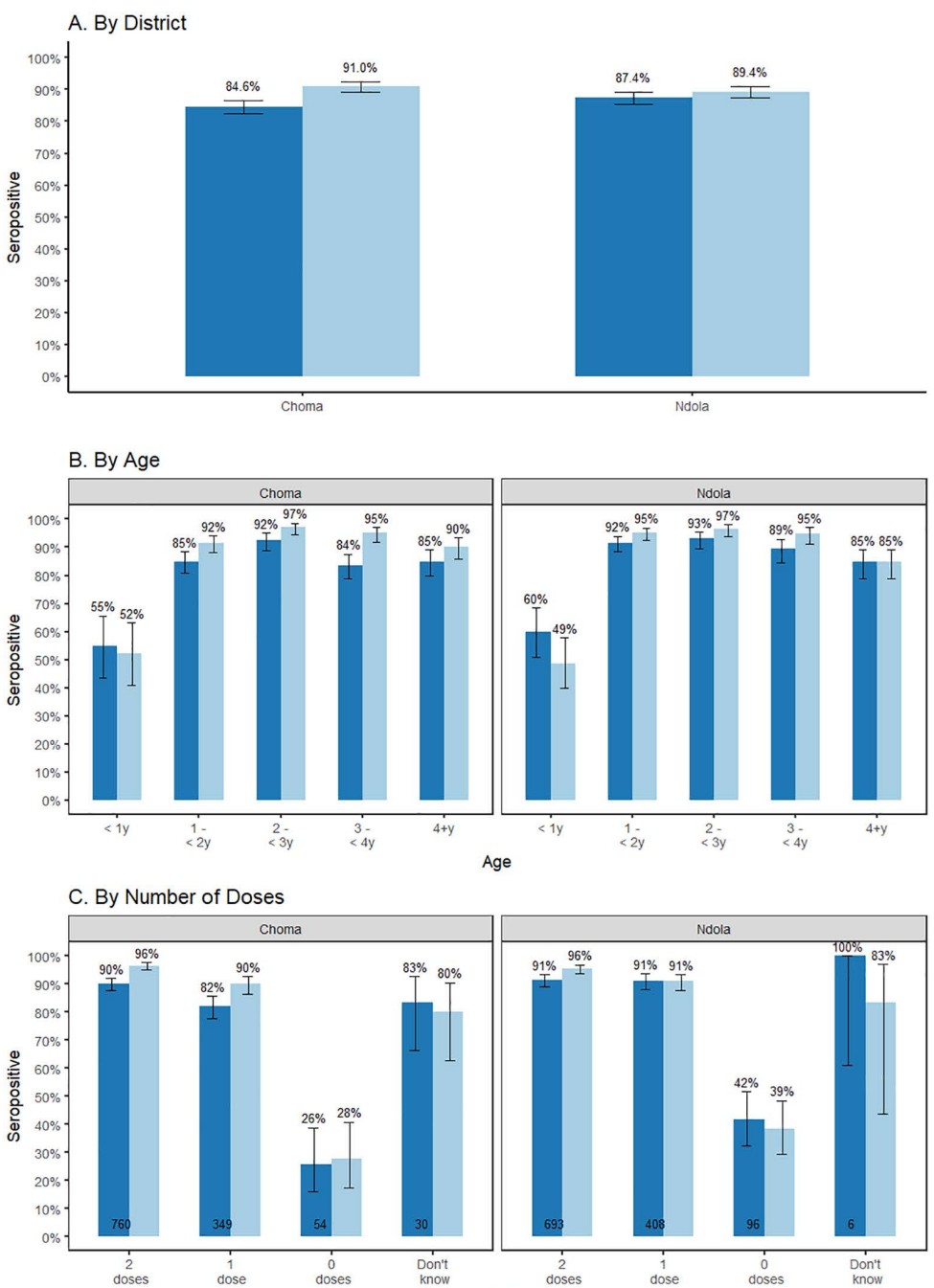

**Fig 4. Measles and rubella seropositivity by district, age, and number of MR doses received, November 2020, Zambia.** We estimated measles and rubella seropositivity among children attending the MR SIA in select facilities in Ndola and Choma Districts. Analyses were stratified by district then further stratified by age of the child and number of MR doses received prior to the SIA.

(95% CI: 87–90%) among those older than 2 years (Fig 4B). In Ndola District among children eligible for vaccination prior to the SIA (≥ 1 year) there was lower rubella seroprevalence among children older than 4 years.

As expected, measles and rubella seroprevalence were significantly associated with the number of MR doses received prior to the SIA (p<0.001) (Fig 4C). Measles and rubella seroprevalence were similar between those with unknown vaccination status compared to those with at least one MR vaccine dose. Findings were similar when stratified by card availability (S5 Fig), although measles and rubella seroprevalence were lower among those for whom vaccination status was based on recall. Thirty-six percent of children with no MR doses were seropositive for measles and 35% for rubella. Of the 57 children with no MR doses who were seropositive for either measles or rubella, 49 were seropositive for both measles and rubella, suggesting their vaccination status was misclassified. Nine percent of children with two doses of MR were seronegative for measles and four percent for rubella.

There was no difference in seroprevalence by rural versus urban setting, type of SIA site, or day of the SIA (S6 Fig). Children with no prior receipt of BCG vaccine and those from families with more children under 5 years of age (Choma District only) were more likely to be measles seronegative (S3 Table).

## Discussion

Supplementary immunization activities, often in the form of weeklong vaccination campaigns, are a common and valuable tool in efforts to reach and maintain immunity levels needed to control and eliminate measles, rubella, and other vaccine preventable diseases. Countries around the world rely on regular mass vaccination campaigns to vaccinate individuals, typically children, who either never received their recommended routine vaccination or who remain susceptible despite previous vaccination [20]. However, despite reliance on these costly activities, it remains largely unknown just how effective and efficient mass vaccination campaigns are in different settings in reaching those children who need immunization, and not just revaccinating children who are already immune. This is particularly critical for measles, for which upward of 95% of the population needs to be immune to achieve and sustain elimination [21].

To quantify this effectiveness and efficiency, we conducted a serological survey nested within a measles and rubella SIA in two districts in Zambia in November 2020. Measles and rubella serological surveys conducted among participants in a mass vaccination campaign provide operationally relevant insights into the vaccination status and serostatus of children at the time of vaccination during the campaign.

The SIA reached a considerable number of un- and under-vaccinated children, including 28% of enrolled children who received an MR vaccine dose they probably would not have otherwise received. However, children vaccinated during this SIA were overall highly likely to already have been vaccinated through the routine immunization system. While this is not surprising, it speaks to the need for more efficient approaches, such as targeted subnational campaigns or differentiated distribution of personnel and resources based on the local context, particularly for countries with high vaccination coverage with two doses of MCV like Zambia. Only 6% of children reached by the campaign were previously unvaccinated and, after adjusting for age and the probability of eventually receiving a routine dose, this drops to only 5%. While SIAs may not always be the most efficient strategy to reach unvaccinated children, they can be effective mechanisms to provide a second vaccination opportunity. For this 2020 SIA in Zambia, effectiveness and efficiency were mixed: we estimate 23% of doses given provided MR2 to children who would have never received it otherwise. However, with 84% of SIA participants compared to 74% in the general population considered fully vaccinated (2 doses among children 24–35 months), this equates to children who already received MR2 being twice as likely to attend the SIA as those who were undervaccinated (OR = 1.84). Serology conducted during the campaign further supported this observation, with most study children measles seropositive by 1 year of age (88%). Among those 2 years and older, children with fewer potential opportunities to be reached by the routine immunization system, we estimated 25 children needed to be vaccinated for every 1 zero-dose child and 7 children needed to be vaccinated for every under-vaccinated child [22].

Despite these inefficiencies, 14% of the children reached by the SIA were measles seronegative, including some who had been previously vaccinated, demonstrating the continued benefits of campaigns at helping to close the population immunity gap by reaching susceptible children. Assuming 85% vaccine efficacy for 1 dose, and 95% for 2 doses [3–5], we would have expected 86% of children younger than five years to be seropositive for measles at the time of the SIA, aligning with the observed seroprevalence. However, these results also demonstrate the shortcomings of basing susceptibility – and thereby risk estimates or vaccination need decisions – purely on vaccination coverage data. In contrast to the 86% seropositivity, 94% of children reported a previous dose of MR1. Further, 9% of the measles seronegative children reported having previously received two doses of measles vaccine, higher than the expected 95% given measles vaccine immunogenicity.

This study demonstrates the information that can be gained rapidly about the effectiveness of an SIA. By nesting a serosurvey in the SIA we were able to quantify the impact of the SIA for reaching under- and unvaccinated children as well as seronegative children. Given the common long delays to perform and receive data from post-campaign coverage surveys, data collected during the campaign can be valuable to track whether the campaign is effectively reaching those for whom it is needed. Further, data on age, vaccination status, and serostatus of children collected at the time of campaigns or other supplemental immunization activities can provide useful insight into the relative efficiency of the different vaccination strategies and how efficiency may vary across settings.

This study sheds light on the characteristics most associated with being un- or under-vaccinated in these districts in Zambia. Several expected characteristics were confirmed in this population. As previously shown, lower maternal education was a strong indicator of lack of prior measles and rubella vaccination [23]. Additionally, children with multiple young siblings were also more likely to miss their second dose, indicating a high drop out among children from larger families. This flags the need to develop strategies to target larger families. These inequities in vaccination status were also identified at the national level [15].

Although we were unable to detect statistical differences in maternal and child characteristics by day of campaign, there were suggestive differences. For example, outreach sites reached more seronegative children towards the end of the campaign. The middle days of the campaign reached the most children of mothers with lower educational levels. Variability by campaign day is not typically reported but could inform the most effective days or ways to reach vulnerable children.

Zambia introduced rubella-containing vaccine during the prior nationwide SIA in 2016 [24]. The 90% rubella seropositivity is evidence of the MR1 coverage having remained above 90% and MR2 coverage increasing from 64% in 2017 to 81% in 2021 [19]. If national estimates are similar, this would be sufficiently high coverage to maintain rubella incidence at low levels and provide an opportunity for rubella elimination in Zambia. The lower rubella seroprevalence among children 4 years of age observed in Ndola District may reflect children who were missed by rubella-containing vaccine during the 2016 MR campaign.

While this study contributes to our understanding of measles vaccination status and seroprevalence among children attending an MR SIA in Zambia, these findings should be considered with several limitations. First, while we have substantial data for children sampled during the SIA, we do not have any information on those children who did not participate in the SIA. This is an important limitation because our estimates of efficiency are limited to extrapolations to overall coverage estimates for Zambia as a whole and may not fully reflect these locations or this time. As such, these seroprevalence estimates may not be representative of the general population. Further, as we only enrolled a relatively small sample of the children attending the campaign, it is also possible that the study children do not fully represent all children who attended the SIA in these districts. Finally, this study was nested in a vaccination campaign that occurred during the COVID-19 pandemic and had lower coverage compared to previous SIAs, which could impact who was included in our study [15]. However, we expect that the potential bias is relatively small in terms of who was missed from the SIA. Measles seroprevalence was higher than in a representative community study in Southern Province prior to the 2016 SIA

(78%) suggesting either routine immunization has improved or the population attending the SIA is more likely to be vaccinated than the general population [25].

This study provides valuable insights into the effectiveness and efficiency of the 2020 MR SIA in Zambia. Monitoring SIA effectiveness, efficiency, and equity is important to understand the benefits of vaccine delivery strategies in reaching zero-dose and under-vaccinated children. Countries like Zambia with high MR1 and MR2 coverage could consider tailored or targeted SIAs to better reach un- and under-vaccination children.

## Supporting information

**S1 Table. Characteristics associated with not receiving measles-rubella vaccine prior to the SIA.** The analysis was restricted to children 12 months and older. The outcome was no MR doses prior to the SIA. Univariable ORs were adjusted for age in years. Analysis with SIA site type (outreach vs fixed) was restricted to health facilities with both fixed and outreach locations. Bold indicates p < 0.05.
(DOCX)

**S2 Table. Characteristics associated with missing the second measles-rubella vaccine dose prior to the SIA among those with at least 1 dose.** This analysis was restricted to children 24 months and older with at least 1 MR dose prior to the SIA. The outcome was no receipt of the second MR dose prior to the SIA. Univariable ORs adjusted for age in years. Analysis with SIA site type (outreach vs fixed) was restricted to health facilities with both fixed and outreach locations. Bold indicates p < 0.05.
(DOCX)

**S3 Table. Characteristics associated with measles seronegativity.**
(DOCX)

**S1 Fig. Age at routine measles-rubella vaccine receipt.** Children were excluded from this analysis if vaccine receipt was based on recall, if the dosing date was prior to their date of birth, or if the MR2 receipt date was on or prior to the date of MR1 receipt. Remaining children with a card but no documented evidence of MR were right-censored at 59 months.
(DOCX)

**S2 Fig. Age at vaccination during the SIA for children receiving their first or second measles vaccine dose through the SIA.** Dashed vertical lines represent anticipated age of MR1 and MR2 receipt (MR1, 9 months and MR2, 18 months).
(DOCX)

**S3 Fig. Variability in descriptive characteristics of children by SIA day.**
(DOCX)

**S4 Fig. Variability by day in the percentage of children with no MR doses prior to the SIA by type of SIA site.** Both districts combined. Restricted to health facilities with both fixed and outreach locations and excluding children < 12 months. P-value for variability by day in percent zero-dose from logistic regression model adjusted for age and district: fixed site, 0.60; outreach site, 0.07.
(DOCX)

**S5 Fig. Measles and rubella seropositivity by number of doses and card availability.**
(DOCX)

**S6 Fig. Measles and rubella seropositivity by setting, SIA type, and campaign day.** Analysis with SIA site type (outreach vs fixed) restricted to health facilities with both fixed and outreach locations.
(DOCX)

**S1 Text. Supplemental methods.**
(DOCX)

**S1 Checklist. Inclusivity in global research.**
(DOCX)

## Acknowledgments

We would like to acknowledge all the children and their parents who agreed to take part in this study. We would also like to acknowledge the Ministry of Health and all the serosurvey staff, vaccinators, nurse in-charges, and district health staff who contributed to the successful implementation of the project. Finally, we acknowledge Monica Pilewskie for her support coding the transcripts from the interviews.

## Author contributions

**Conceptualization:** Christine Prosperi, Shaun Truelove, Andrea C. Carcelen, Gershom Chongwe, Francis D. Mwansa, Phillimon Ndubani, Edgar Simulundu, Amy K. Winter, William J. Moss, Simon Mutembo.

**Data curation:** Shaun Truelove.

**Formal analysis:** Christine Prosperi, Shaun Truelove, Andrea C. Carcelen, Amy K. Winter.

**Funding acquisition:** Shaun Truelove, William J. Moss.

**Investigation:** Christine Prosperi, Shaun Truelove, Andrea C. Carcelen, Innocent C. Bwalya, Mutinta Hamahuwa, Kelvin Kapungu, Kalumbu H. Matakala, Gloria Musukwa, Irene Mutale, Evans Betha, Nchimunya Chaavwa, Lombe Kampamba, Japhet Matoba, Passwell Munachoonga, Webster Mufwambi, Ken Situtu, Amy K. Winter, Simon Mutembo.

**Methodology:** Christine Prosperi, Shaun Truelove, Andrea C. Carcelen, Amy K. Winter, William J. Moss.

**Project administration:** Gershom Chongwe, Francis D. Mwansa, Phillimon Ndubani, Edgar Simulundu, Philip E. Thuma, William J. Moss, Simon Mutembo.

**Resources:** Innocent C. Bwalya, Mutinta Hamahuwa, Kelvin Kapungu, Kalumbu H. Matakala, Gloria Musukwa, Irene Mutale, Evans Betha, Nchimunya Chaavwa, Lombe Kampamba, Japhet Matoba, Passwell Munachoonga, Webster Mufwambi, Ken Situtu, Philip E. Thuma, Constance Sakala, Princess Kayeye.

**Software:** Shaun Truelove.

**Supervision:** Shaun Truelove, Gershom Chongwe, Francis D. Mwansa, Phillimon Ndubani, Edgar Simulundu, Philip E. Thuma, William J. Moss, Simon Mutembo.

**Validation:** Shaun Truelove.

**Visualization:** Shaun Truelove.

**Writing – original draft:** Christine Prosperi, Shaun Truelove, Andrea C. Carcelen.

**Writing – review & editing:** Shaun Truelove, Gershom Chongwe, Francis D. Mwansa, Phillimon Ndubani, Edgar Simulundu, Innocent C. Bwalya, Mutinta Hamahuwa, Kelvin Kapungu, Kalumbu H. Matakala, Gloria Musukwa, Evans Betha, Nchimunya Chaavwa, Lombe Kampamba, Japhet Matoba, Passwell Munachoonga, Webster Mufwambi, Ken Situtu, Philip E. Thuma, Constance Sakala, Princess Kayeye, Amy K. Winter, Matthew J Ferrari, William J. Moss, Simon Mutembo.

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
