## [Decision Letter · Decision Letter 0]

13 Jun 2024

PGPH-D-24-00874

High measles and rubella vaccine coverage and seroprevalence among Zambian children participating in a measles and rubella supplementary immunization activity

Dear Dr. Truelove,

Thank you for submitting your manuscript to PLOS Global Public Health. After careful consideration, we feel that it has merit but does not fully meet PLOS Global Public Health’s publication criteria as it currently stands. Therefore, we invite you to submit a revised version of the manuscript that addresses the points raised during the review process.

We look forward to receiving your revised manuscript.

Kind regards,

Abraham D. Flaxman, Ph.D.

Academic Editor

Journal Requirements:

2. Please include a complete copy of PLOS’ questionnaire on inclusivity in global research in your revised manuscript. Our policy for research in this area aims to improve transparency in the reporting of research performed outside of researchers’ own country or community. The policy applies to researchers who have travelled to a different country to conduct research, research with Indigenous populations or their lands, and research on cultural artefacts. The questionnaire can also be requested at the journal’s discretion for any other submissions, even if these conditions are not met.  Please find more information on the policy and a link to download a blank copy of the questionnaire here: https://journals.plos.org/globalpublichealth/s/best-practices-in-research-reporting. Please upload a completed version of your questionnaire as Supporting Information when you resubmit your manuscript.

3. Please amend your detailed Financial Disclosure statement. This is published with the article. It must therefore be completed in full sentences and contain the exact wording you wish to be published.

Additional Editor Comments (if provided):

Thank you for submitting this interesting study. You will see that one reviewer has a number of suggestions on how to make the methods clearer. I would like to make sure that this reviewer has another chance to consider your results and conclusion after you have clarified the methods as much as possible.

Reviewers' comments:

Reviewer's Responses to Questions

**Comments to the Author**

1. Does this manuscript meet PLOS Global Public Health’s publication criteria ? Is the manuscript technically sound, and do the data support the conclusions? The manuscript must describe methodologically and ethically rigorous research with conclusions that are appropriately drawn based on the data presented.

Reviewer #1: Yes

Reviewer #2: Yes

2. Has the statistical analysis been performed appropriately and rigorously?

Reviewer #1: N/A

Reviewer #2: Yes

3. Have the authors made all data underlying the findings in their manuscript fully available (please refer to the Data Availability Statement at the start of the manuscript PDF file)?

Reviewer #1: Yes

Reviewer #2: Yes

4. Is the manuscript presented in an intelligible fashion and written in standard English?

Reviewer #1: Yes

Reviewer #2: Yes

5. Review Comments to the Author

Reviewer #1: Abstract

The method section of the abstract discusses only data collection. I think it would be beneficial for readers to have a summary of the methods used. In the results section of the abstract, the authors provide an estimate for the percentage of children that wouldn’t have received MR dose 1 or MR dose 2. I almost always find it peculiar when people use the term "estimate" without showing the uncertainty levels. Of course, there is a time and place for everything, and this seems like the right one.

Introduction

Very well written and succinct. Just one comment: I believe paragraph two can benefit from more references.

Methods:

Survey setting- This paragraph is not entirely clear to me, “For facilities with multiple outreach sites, the survey teams moved with the vaccination team to the different locations where possible; otherwise, sites expected to have the most children during the campaign were selected.” Overall, how frequently were these decisions made?

Statistical analysis: First of all, however simple or complex the model one is using, one must always include mathematical/statistical formulas. For one, it makes understanding easier and, most importantly, it reassures that the authors have fully understood the implications and the use of the model/method.

Second, I am not exactly sure if the comparison outlined in this paragraph is a fair comparison: “Estimates of measles-rubella vaccine dose 1 (MR1) and dose 2 (MR2) vaccination status among children 12 to 23 months of age and 24 to 35 months of age, respectively, were compared with 2020 estimates from WHO-UNICEF (WUENIC).”

Third, the authors have indicated that the association between vaccination status and day of SIA and other child-level characteristics were explored with logistic regression. For me, the use of the term “other child-level characteristics” is a bit concerning. What are these “others”? If included, please point me to the right place in the paper. Furthermore, the authors implied they used predicted probabilities for MR1 at 12 months or MR2 at 24 months. There are many ways to construct probabilities; it's not clear what has been used. Moreover, the statement, “Predicted probability of MR1 at 12 months, or MR2 at 24 months, was estimated for each of the 30 SIA sites using district-specific logistic regression models adjusted for age in years to account for differences in age distribution by SIA site,” seems very vague as far as models go. Clear specifications of the model(s) and rationale (I mean substantive ones), for choosing this model, are needed. The same comment goes for the logistic regression model used to study association between measles seropositive and child-level characteristics. My argument is that with proper substantive justification, one can always achieve a higher level of rigor using more advanced model setup (not always, the simpler ones can be well justified as well).

Effectiveness and efficiency of vaccination activities: In general, I remain a bit skeptical about how the effectiveness of vaccination activity was measured, especially whether the Odds Ratio (OR) comparing the SIA values with the general population is robust, given the many biases at play here (from both). A strong justification for the validity of this method might be required from the authors. The same thing also applies to how efficacy was estimated; in addition, for efficacy, the use of the 2018 Demographic and Health Survey and the 2020 SIA was not fully described (temporal variation). Moreover, personally, for the objective or measurement the authors are trying to make here, a more nomenclature, which is not so broad and general, and which is very specific, can be used.

Results:

For the results section, I will only have one question (when I have more clarity and more justification for choices of methods in the method section, I can get back to this); however, the 75% and 91% agreement for participation in the serosurvey study across the two districts is significant. I wonder if there are some theories behind the differences between these two?

Reviewer #2: It would have been good if the authors had briefly written about the selection criteria of the study sites, which would have given much more clarity in selecting the study sites. With this modification the manuscript may be accepted for publication.

6. PLOS authors have the option to publish the peer review history of their article (what does this mean? ). If published, this will include your full peer review and any attached files.

**Do you want your identity to be public for this peer review?** For information about this choice, including consent withdrawal, please see our Privacy Policy .

Reviewer #1: **Yes: ** Latera Olana

Reviewer #2: No

---

## [Decision Letter · Decision Letter 1]

25 Sep 2024

PGPH-D-24-00874R1

High measles and rubella vaccine coverage and seroprevalence among Zambian children participating in a measles and rubella supplementary immunization activity

Dear Dr. Truelove,

Thank you for submitting your manuscript to PLOS Global Public Health. After careful consideration, we feel that it has merit but does not fully meet PLOS Global Public Health’s publication criteria as it currently stands. Therefore, we invite you to submit a revised version of the manuscript that addresses the points raised during the review process.

You will see that Reviewer 1 came up with a number of additional minor comments while going through your revision, and I think that it is worthwhile for you to do another round of revisions to benefit from the possibility of increasing clarity (and therefore readership) based on this.

Please submit your revised manuscript by . If you will need more time than this to complete your revisions, please reply to this message or contact the journal office at globalpubhealth@plos.org. Please include the following items when submitting your revised manuscript:

We look forward to receiving your revised manuscript.

Kind regards,

Abraham D. Flaxman, Ph.D.

Academic Editor

Journal Requirements:

Additional Editor Comments (if provided):

Reviewers' comments:

Reviewer's Responses to Questions

**Comments to the Author**

1. If the authors have adequately addressed your comments raised in a previous round of review and you feel that this manuscript is now acceptable for publication, you may indicate that here to bypass the “Comments to the Author” section, enter your conflict of interest statement in the “Confidential to Editor” section, and submit your "Accept" recommendation.

Reviewer #1: All comments have been addressed

2. Does this manuscript meet PLOS Global Public Health’s publication criteria ? Is the manuscript technically sound, and do the data support the conclusions? The manuscript must describe methodologically and ethically rigorous research with conclusions that are appropriately drawn based on the data presented.

Reviewer #1: Yes

3. Has the statistical analysis been performed appropriately and rigorously?

Reviewer #1: Yes

4. Have the authors made all data underlying the findings in their manuscript fully available (please refer to the Data Availability Statement at the start of the manuscript PDF file)?

Reviewer #1: (No Response)

5. Is the manuscript presented in an intelligible fashion and written in standard English?

Reviewer #1: Yes

6. Review Comments to the Author

Reviewer #1: Abstract: Minor Comments

1. Clarity: “We used the expected vaccination probability by age derived from Demographic and Health Surveys to quantify the value of the immunization campaign.” Can the author use more descriptive words instead of “value”?

Introduction: Minor Comments

1. In paragraph 2, “...but their results are often subject to bias, substantial uncertainty, and...” What kind of bias or uncertainty are the authors implying? Be as clear and specific as possible. Given the objective of the paper, shouldn't the description of the gaps in post-campaign surveys focus on the fact that they cannot answer how many susceptible (as defined by the paper) children received the vaccine?

2. In the same paragraph, “Understanding whether susceptible...” could be improved by using more objective statements like “estimating” instead of “understanding.”

3. “Zambia has a strong childhood vaccination program with 93% coverage in 2019 for the first dose of the measles-rubella (MR) vaccine but only 66% for the second dose.” Can this be phrased differently? The first part of the sentence and the last part seem conflicting.

4. In the same paragraph, the authors indicated that the 3 million doses delivered during Child Health Week might reflect suboptimal coverage. How does this 3 million translate into coverage? Would it be suboptimal? If not, could COVID-19 and operational challenges explain the subsequent lack of protection for outbreaks?

Materials and Methods: Minor Comments

1. The first paragraph of the method section may need a reference.

2. “These characteristics included geographic location and setting (urban versus rural), type of health facility, size of the health facility catchment area, functionality of the health facility based on national performance metrics, and details on accessibility of the health facility or difficult-to-reach subpopulations in the catchment area.” Were all these details acquired from site staff? If so, I am just thinking of the different limitations the authors can include in their discussion section.

3. If I missed it, I apologize, but is the travel time data from the participants (caregivers) or from site staff? In both cases there will be a limitation worth acknowledging, as objective measurement would require considering factors like different terrains, transportation availability, and other logistical elements.

4. “Five facilities in each district were selected to have serosurvey enrollment at both fixed and outreach posts, for a total of 30 campaign sites between the two districts. Outreach sites expected to have the most children during the campaign were selected.” This statement is repetitive; the previous sentence already conveys the same information.

5. In the last paragraph of the Specimen collection and testing section, “A stratified random subsample of 300 DBS specimens was selected from those collected in Ndola District.” Why only Ndola District (just wanted to know)?

6. In this same paragraph there is an abbreviation for EIA, which was never defined before.

7. In the statistical analysis section, the authors specified they used the vaccination card as one variable in their logistic regression model for the outcome, vaccination status. Could you provide a brief description of the perceived association between these two and the need for controlling this especially related to other related variables that the authors already controlled for (just for my understanding, no need to change anything)?

Results: Minor Comments

1. Table 1 doesn’t contain all the variables used in the model. Can it be made more comprehensive? (I am assuming this table was intended for the variables used in the main analysis. If not, and the choice of what was included is intentional, feel free to ignore this comment.)

2. Maternal education, looking at secondary or higher, is highest in Ndola, which aligns with expectations given that Ndola is an urban district. However, for primary or less, Choma is higher, and I believe much of the percentage in this category comes from individuals with less than primary education, which I expect to be higher in Choma. If the second category were changed from “primary or less” to “primary” as one group and “less than primary” as a separate group, the percentage distribution might change and this might have significant impact on the significance level and estimated coefficient of the model.

3. In the Vaccination status section: “(n=2,364; 98.5%), 94% (n=2,214).” The author should ensure consistency in the writing format.

4. In paragraph 3 of the results section, the authors use “vaccination activity effectiveness.” They should consistently use the terminology they coined throughout the paper (VAET).

5. In paragraph 3 of the results, the authors refer to “Lower maternal education.” Does this refer to primary or less education?

6. In the Measles and rubella seroprevalence section, the term “two sites” is used. The term "site" has been inconsistently applied throughout the paper—sometimes referring to hospitals (30 SIA sites) and now to districts. It would help to maintain consistency.

General Comment

1. For anyone who is skeptical of the methods used, how can the authors demonstrate beyond reasonable doubt that the methods throughout the paper are robust? This could involve considering sensitivity analyses, changing some assumptions, or re-evaluating how the unknown status was treated. Obviously, there are so many ways to show robustness.

7. PLOS authors have the option to publish the peer review history of their article (what does this mean? ). If published, this will include your full peer review and any attached files.

**Do you want your identity to be public for this peer review?** For information about this choice, including consent withdrawal, please see our Privacy Policy .

Reviewer #1: **Yes: ** Latera Tesfaye Olana

---

## [Decision Letter · Decision Letter 2]

26 Apr 2025

PGPH-D-24-00874R2

High measles and rubella vaccine coverage and seroprevalence among Zambian children participating in a measles and rubella supplementary immunization activity

Dear Dr. Truelove,

Thank you for submitting your manuscript to PLOS Global Public Health. After careful consideration, we feel that it has merit but does not fully meet PLOS Global Public Health’s publication criteria as it currently stands. Therefore, we invite you to submit a revised version of the manuscript that addresses the points raised during the review process.

the revised manuscript has been reviewed and some additional comments provided by the reviewer. Please review their assessment and make the appropriate revisions to improve the manuscript. 

We look forward to receiving your revised manuscript.

Kind regards,

Emma Campbell, Ph.D

Staff Editor

Journal Requirements:

Reviewers' comments:

Reviewer's Responses to Questions

**Comments to the Author**

1. If the authors have adequately addressed your comments raised in a previous round of review and you feel that this manuscript is now acceptable for publication, you may indicate that here to bypass the “Comments to the Author” section, enter your conflict of interest statement in the “Confidential to Editor” section, and submit your "Accept" recommendation.

Reviewer #3: (No Response)

2. Does this manuscript meet PLOS Global Public Health’s publication criteria ? Is the manuscript technically sound, and do the data support the conclusions? The manuscript must describe methodologically and ethically rigorous research with conclusions that are appropriately drawn based on the data presented.

Reviewer #3: Yes

3. Has the statistical analysis been performed appropriately and rigorously?

Reviewer #3: I don't know

4. Have the authors made all data underlying the findings in their manuscript fully available (please refer to the Data Availability Statement at the start of the manuscript PDF file)?

Reviewer #3: Yes

5. Is the manuscript presented in an intelligible fashion and written in standard English?

Reviewer #3: Yes

6. Review Comments to the Author

Reviewer #3: This manuscript is well-written and very interesting and will make a good addition to the measles SIA literature.

I have only minor optional suggestions for clarification. The one I feel most strongly about is that the language about regression should clarify whether the regression accounted for the complex sample design of the study. Was it done with something like the ‘survey’ package in R or the svy: prefixes in Stata? If not, I think it should be re-done. All other comments are just suggestions.

1. The estimation of hazard, or the probability that each un/under-vx’d child would or would have not received the dose otherwise is an interesting element of this work. I would like to see in one of the Annexes some documentation of the sample size on which those models were based. How many children age 12-23 and 23-35 were in the DHS dataset that you used to estimate the hazards? It might be helpful to see a table listing number of children by months in age and the % of those that had received MCV1 and MCV2. But at the very least, please report the total N for each cohort and each district.

2. Was there a post-campaign coverage survey for this SIA? If yes, please report the estimated campaign coverage from that work. If no, can you report an estimated coverage from administrative figures? It is one thing to say (line 89) that the campaign reached over 3M children…that sounds good…but for persons with an interest in measles SIAs, we want to judge the work against the benchmark of 95% target coverage. Please share what you know with the reader re: campaign coverage.

3. Line 127. The number 2.1 for DEFF is a curious value. Mention in a footnote, perhaps, how it was calculated.

4. Line 161. Clarify that your logistic regression accounted for the complex sample design. If it did *not* do so, then I strongly suggest that you need to redo it.

5. Line 164. I applaud you for listing what kind of CI you calculated. Wald is not a good choice if the estimated prevalence approaches 0% or 100%. (See Dean & Pagano 2015 and many of the papers they cite.) I don’t require that you change it here, but for your future reference, I don’t think there is ever a good reason to ever use a Wald interval for a proportion. The better intervals like logit or Wilson or others approximate the Wald when the sample size is high or the outcome is near 50% and they only get better as the estimate goes to the extremes. Some options, like Clopper-Pearson are too conservative for usual use, but you’ll never go wrong using a Wilson interval which is also defined when the coverage is 0% or 100% (which some, like logit, are not.)

Dean, Natalie, and Marcello Pagano. 2015. “Evaluating Confidence Interval Methods for Binomial Proportions in Clustered Surveys.” Journal of Survey Statistics and Methodology 3 (4): 484–503. https://doi.org/10/gnjws5.

6. Effectiveness and efficiency: People have a difficult time thinking about odds ratios, so I’m not optimistic that these proposed measures will be widely used, but you have explained them clearly enough to use here.

7. Line 257. A point estimate of 39% probably shouldn’t have a 95% CI of 34-40%. Check the numbers. You said you were using symmetric Wald intervals. And even the asymmetric intervals all have the “long” tail pointing toward 50%...not pointing away from 50%.

8. General comment: I like that many of your proportions omit the decimal place and any digits afterward. I think that improves readability. I find it districting when you switch back and forth between showing the decimal digit and not, as in the paragraph from lines 255-268.

9. Under limitations, mention that purposeful site selection is another reason that the sample may not be representative of all SIA participants.

10. Under limitations, mention that variance calculations and confidence intervals in things like the regression analysis assume that the data are a representative sample; because this design used purposeful selection of data collection sites, we might think of the associations and outcomes here as being indicative but not representative.

11. I recommend adding one or two more sentences about the children with no history of immunization whose serology indicated immunity to measles. These might be considered “previously zero-dose” children who did not need, in some sense, to be vaccinated as they were already protected. Is that presumably because they had contracted the disease when younger? This category of children is not commonly discussed in literature assessing the benefits of SIAs. Would there have already been an estimate of the % of zero-dose children who were immune to measles in Zambia? The serology is what could make a precise estimate of the prevalence known if the sample were random. Without serology, I suppose a PCCS could ask the caregiver if the child ever experienced measles. I know some PCCSs have asked that in Ethiopia, for instance, but of course many caregivers will not be able to report that detail of their child’s medical history accurately. I think those children warrant a few more sentences to describe how that outcome comes about and to say that perhaps they might not have been otherwise vaccinated, and so counted among the successes of the campaign, but they didn’t really need to be vaccinated, either.

7. PLOS authors have the option to publish the peer review history of their article (what does this mean? ). If published, this will include your full peer review and any attached files.

**Do you want your identity to be public for this peer review?** For information about this choice, including consent withdrawal, please see our Privacy Policy .

Reviewer #3: No

---

## [Decision Letter · Decision Letter 3]

11 Aug 2025

High measles and rubella vaccine coverage and seroprevalence among Zambian children participating in a measles and rubella supplementary immunization activity

PGPH-D-24-00874R3

Dear Dr. Truelove,

We are pleased to inform you that your manuscript 'High measles and rubella vaccine coverage and seroprevalence among Zambian children participating in a measles and rubella supplementary immunization activity' has been provisionally accepted for publication in PLOS Global Public Health.

Best regards,

Mara Jana Broadhurst, M.D., Ph.D.

Academic Editor

Reviewer Comments (if any, and for reference):

Reviewer's Responses to Questions

**Comments to the Author**

1. If the authors have adequately addressed your comments raised in a previous round of review and you feel that this manuscript is now acceptable for publication, you may indicate that here to bypass the “Comments to the Author” section, enter your conflict of interest statement in the “Confidential to Editor” section, and submit your "Accept" recommendation.

Reviewer #3: All comments have been addressed

2. Does this manuscript meet PLOS Global Public Health’s publication criteria ? Is the manuscript technically sound, and do the data support the conclusions? The manuscript must describe methodologically and ethically rigorous research with conclusions that are appropriately drawn based on the data presented.

Reviewer #3: Yes

3. Has the statistical analysis been performed appropriately and rigorously?

Reviewer #3: Yes

4. Have the authors made all data underlying the findings in their manuscript fully available (please refer to the Data Availability Statement at the start of the manuscript PDF file)?

Reviewer #3: Yes

5. Is the manuscript presented in an intelligible fashion and written in standard English?

Reviewer #3: Yes

6. Review Comments to the Author

Reviewer #3: Thank you for the clear responses and for the clarifications in the manuscript.

7. PLOS authors have the option to publish the peer review history of their article (what does this mean? ). If published, this will include your full peer review and any attached files.

**Do you want your identity to be public for this peer review?** For information about this choice, including consent withdrawal, please see our Privacy Policy .

Reviewer #3: No
